# Towards a Collective Medical Imaging AI: Enabling Continual Learning from Peers

**Guangyao Zheng**[1]                                                          TZ30@RICE.EDU

**Vladimir Braverman**[1]                                                      VB21@RICE.EDU

**Jeffrey Leal** [3]                                                           JLEAL1@JHMI.EDU

**Steven Rowe** [7]                                                     STEVEN_ROWE@MED.UNC.EDU

**Doris Leung** [5,6]                                                   LEUNGD@KENNEDYKRIEGER.ORG

**Michael A. Jacobs**[2,3]                                              MICHAEL.A.JACOBS@UTH.TMC.EDU

**Vishwa S. Parekh**[4]                                                 VPAREKH@SOM.UMARYLAND.EDU

[1] *Department of Computer Science, Rice University, Houston, TX, USA*

[2] *Department Of Diagnostic And Interventional Imaging, McGovern Medical School, UTHealth Houston, Houston, TX, USA,*

[3] *The Russell H. Morgan Department of Radiology and Radiological Science, The Johns Hopkins University School of Medicine, Baltimore, MD 21205*

[4] *University of Maryland Medical Intelligent Imaging (UM2ii) Center, University of Maryland School of Medicine, Baltimore, MD 21201*

[5] *Kennedy Krieger Institute, 716 North Broadway, Room 411, Baltimore, MD 21205*

[6] *Department of Neurology, Johns Hopkins University School of Medicine, Baltimore, MD, USA*

[7] *Department of Radiology, UNC School of Medicine, Chapel Hill, NC*

**Editors:** Accepted for publication at MIDL 2024

## Abstract

Federated learning is an exciting area within machine learning that allows cross-silo training of large-scale machine learning models on disparate or similar tasks in a privacy-preserving manner. However, conventional federated learning frameworks require a synchronous training schedule and are incapable of lifelong learning. To that end, we propose an asynchronous decentralized federated lifelong learning (ADFLL) method that allows agents in the system to asynchronously and continually learn from their own previous experiences and others', thus overcoming the potential drawbacks of conventional federated learning. We evaluate the ADFLL framework in two experimental setups for deep reinforcement learning (DRL) based landmark localization across different imaging modalities, orientations, and sequences. The ADFLL was compared to central aggregation and conventional lifelong learning for upper-bound comparison and with a conventional DRL model for lower-bound comparison. Across all the experiments, ADFLL demonstrated excellent capability to collaboratively learn all tasks across all the agents compared to the baseline models in in-distribution and out-of-distribution test sets. In conclusion, we provide a flexible, efficient, and robust federated lifelong learning framework that can be deployed in real-world applications.

**Keywords:** Federated learning, Lifelong learning, Deep reinforcement learning, Landmark localization

## 1. Introduction

Federated Learning (FL) has recently emerged as a promising approach enabling multiple agents to train a model collaboratively without sharing their data (Adnan et al., 2022), which protects data privacy and reduces computational costs at the local agent level by distributing the computation to multiple agents to train the model on their local data and sharing only the model updates with a central server. FL implementations have shown promising results in various medical applications (Roth et al., 2020; Jiang et al., 2022; Yan et al., 2021). FL frameworks often rely on synchronized learning schedules, meaning all participating agents train simultaneously and synchronously communicate with the server. Additionally, conventional FL approaches cannot perform Lifelong Learning (LL) to dynamically integrate information from newer nodes joining the federation or retain information from older nodes leaving the federation, an essential aspect of machine learning applied to medical imaging with constantly changing medical imaging protocols and emergence of new diseases such as COVID-19 (Karani et al., 2018a; Zheng et al., 2023; Derakhshani et al., 2022; Karani et al., 2018b).

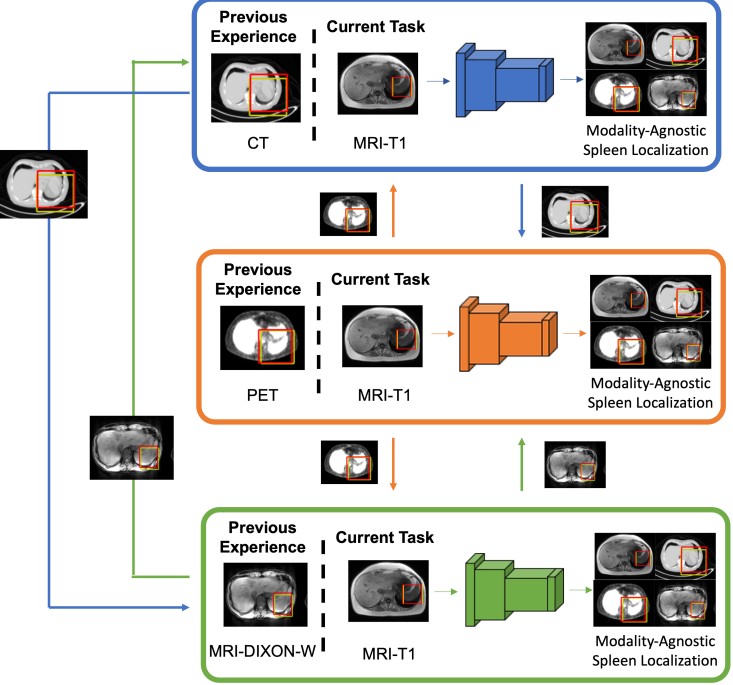

Figure 1: Illustration of asynchronous decentralized federated lifelong learning (ADFLL) set up for cross-modality 3D localization of spleen. The blue, orange, and green boxes represent different agents in the setup. Each agent sequentially encounters two different imaging modalities for along with experiences shared by the other nodes, enabling them to learn to localize the spleen across all four modalities (as opposed to just the two they encountered).

To address these limitations, we propose ADFLL, an asynchronous decentralized federated lifelong learning framework constructed using a network of lifelong learning nodes. We develop and evaluate ADFLL using deep reinforcement learning (DRL) as the base model architecture at each node for the task of 3D landmark localization in medical imaging volumes. DRL with LL capability using selective experience replay (SERIL) has previously been validated for continual landmark localization across different imaging environments using DRL for medical imaging and also forms the basis of LL for each node in ADFLL (Isele and Cosgun, 2018; Rolnick et al., 2019; Zheng et al., 2023). SERIL techniques allow for LL by simultaneously training from a selected sample of a model's previous experiences in addition to the model's current task. ADFLL expands this capability of SERIL to a collaborative setup by allowing SERIL nodes to not only learn from their own previous experiences but also from experiences shared by collaborator nodes, thereby enabling continual learning from peers, as shown in Figure 1.

In this work, we evaluated ADFLL across two different experimental setups for cross-domain 3D landmark localization in multiparametric brain MRI and multi-modality whole body imaging in-distribution and out-of-distribution datasets using deep reinforcement learning (DRL). In addition, we evaluated the ability of ADFLL framework to scale with increase number of agents as well as integrate and retain information without catastrophic forgetting in different LL scenarios with nodes leaving and joining the framework.

## 2. Method

### 2.1. Lifelong deep reinforcement learning

We implemented DRL based on the deep Q-network (DQN) algorithm, as illustrated in Figure 2. The 3D DQN model implemented in this work was adapted from (Mnih et al., 2013; Alansary et al., 2018; Vlontzos et al., 2019; Parekh et al., 2020). The environment was depicted using a 3D imaging volume and the agent was represented by a 3D bounding box capable of six distinct actions: $a \in$ x++, x--, y++, y--, z++, z--. The state was defined by an agent's current location (or a chain of locations), each represented by a 3D bounding box of $45 \times 45 \times 11$ pixels. The reward is calculated by the change in distance to the target landmark location before and after the agent takes an action. The agent's exploration within the environment generated state-action-reward-resulting state $[s, a, r, s']$ tuples, which are recorded and sampled in the experience replay buffer (ERB) over multiple episodes.

We implemented lifelong learning using selective experience replay (Rolnick et al., 2019; Zheng et al., 2023), a model-agnostic lifelong learning approach that enables sharing experiences across different models. ERBs produced by DRL agents during the previous training enable us to achieve lifelong learning. To learn a generalized representation of current and past tasks, the model selects a batch of experiences from the ERB of its current task and the ERBs of previous tasks during training. The information in the ERBs is non-sensitive as the state and resulting states are tiny fractions of the total 3D image, roughly 0.3%.

### 2.2. Asynchronous Decentralized Federated Lifelong Learning

We developed the Asynchronous Decentralized Federated Lifelong Learning (ADFLL) by constructing a network of lifelong DRL agents with the additional modification in the train-

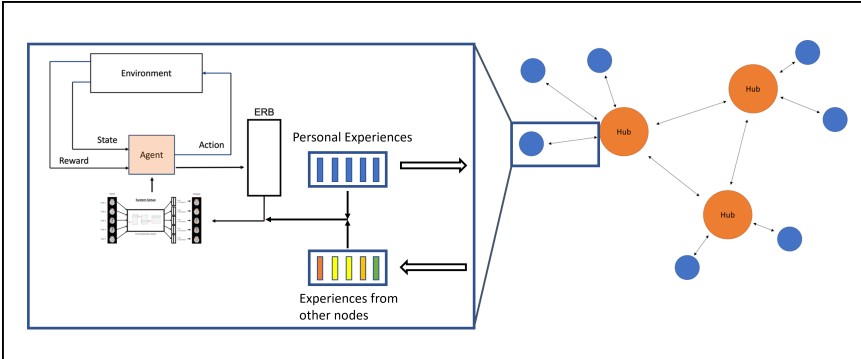

Figure 2: Illustration of asynchronous decentralized federated lifelong learning setup. Blue circles represent individual agents, and orange circles represent the hubs. Each agent is capable of training from current task ERBs, personal experience ERBs, and ERBs from other agents.

ing setup that allows each agent to sample experiences from the current dataset ERB, the agent's personal experiences, and the incoming experiences from other agents, as shown in Figure 2. Each agent shares their database of personal experiences with each other to facilitate learning from each other's experiences. More specifically, once an agent finishes training with a dataset, the resulting experience from the training is shared with the network. As a result, every agent in the network can learn from each other's experiences, thereby integrating federated lifelong learning capability.

Furthermore, to address the communication inefficiencies of an all-to-all decentralized federated learning framework, we use a predefined set of hub nodes that act as communication hubs for spatially adjacent nodes in the network, as shown in Figure 2. Subsequently, every node in the network exclusively and asynchronously communicates with its nearest hub node. By periodically synchronizing their databases with each other, the hub nodes regulate and preserve the experiences in the system in case of communication failures or node dropouts. Our code is available at: https://github.com/vishwaparekh/ADFLL.

## 3. Experimentsal Setup

### 3.1. Baseline Agents

**All-knowing agent:** Agent X is trained using central aggregation of all the data across all the ADFLL agents in one place. This gives us the baseline "upper bound" performance for comparison to the ADFLL agents.

**Conventional lifelong DRL agent:** Agent M has sequential access to all the datasets and forms an "upper bound" comparison between lifelong learning with and without experience sharing between lifelong learning DRL agents.

**Conventional DRL agent:** Agent Y is a conventional DRL with access to only one dataset, forming our "lower bound" baseline for performance comparison with ADFLL agents.

### 3.2. Clinical Data

**Multiparametric Brain MRI Dataset:** We used the multiparametric MRI (mpMRI) brain tumor segmentation (BraTS) dataset (Menze et al., 2014) in this study. This dataset consisted of 285 patients and included pre-contrast T1-weight, post-contrast T1-weighted, T2-weighted, and Fluid Attenuated Inversion Recovery (FLAIR) sequences in the axial orientation. We randomly sampled a subset of 100 patients for this study (60 patients with high-grade glioma (HGG) and 40 patients with low-grade glioma (LGG)). We split the 100 patients into two parts 80:20. 80 were used for training and 20 for evaluation, with the training set consisting of 48 HGG and 32 LGG tumors and the test set consisting of 12 HGG and 8 LGG tumors. We reconstructed the dataset to include all three imaging orientations (coronal, sagittal, and axial). As a result, we obtained twenty-four imaging environments with combinations of two pathologies, 4 imaging sequences, and 3 image orientations. All the experiments were performed for the task of left ventricle localization. A sample of 8 task-environment pairs is shown in Appendix Fig. 4.

**Multimodality Whole-Body Imaging Dataset:** All studies were performed in accordance with the institutional guidelines for clinical research under a protocol approved by our Institutional Review Board (IRB), and all HIPAA agreements were followed for this retrospective study. Our dataset consisted of 50 images volumes, 10 from each imaging sequence type (PET, CT, MRI-T1, MRI-DIXON-F, and MRI-DIXON-W). The PET/CT dataset was acquired using a Biograph mCT 128-slice PET/CT scanner (Siemens Healthineers). For the 18F-DCFPyL scans, patients were intravenously injected with no more than 333 MBq (9 mCi) of radiotracer approximately 60 min before image acquisition. The field of view was vertex to mid-thigh for 18F-DCFPyL. The T1, DIXON-F, and DIXON-W sequences were sampled from the whole body (WB) multiparametric MRI dataset that was acquired using an imaging protocol that scanned from the shoulders to the lower mid calf and described in (Leung et al., 2020). Each of the five imaging sequences was annotated with 3D spleen localization for the experiments. In addition thirty-seven imaging volumes were randomly sampled from the same datasets for internal validation (10 PET, 3 CT, 4 MRI-T1, and 10 MRI-DIXON-F and W).

*Test Set:* All the ADFLL agents and the baseline agents were evaluated using external out-of-distribution datasets sourced from The Cancer Imaging Archive (TCIA) and the Medical Segmentation Decathlon (MSD) databases. More specifically, we sampled a total of 10 PET volumes from the NSCLC-Radiogenomics (N=4) and ACRIN-NSCLC-FDG-PET (N=6) collections (Gevaert et al., 2012; Clark et al., 2013; Machtay et al., 2013; Bakr et al., 2017, 2018; Kinahan et al., 2019), 41 CT volumes from the spleen MSD dataset (Antonelli et al., 2022), and 10 WB mpMRI volumes from the CMB-CRC (N=3), CPTAC-CCRCC (N=2), CPTAC-PDA (N=3), and CPTAC-UCEC (N=2) (Clark et al., 2013).

### 3.3. Experiments

**Cross-Environment 3D ventricle localization:** We initialized this experiment with four agents and a sub-sample of eight brain mpMRI environments (shown in Fig. 4). We implemented two agents on an NVIDIA DGX-1, each with an NVIDIA V100, and two on Google Cloud, each with an NVIDIA T4. The two agents on Google Cloud, A1 and A2, have their individual hubs, H1 and H2 whereas the remaining two agents, A3 and A4, on

the DGX-1 were connected to the third hub, H3. We trained all four agents for three iterations. At the beginning of each iteration, each agent received a randomly selected training dataset with a different environment. As a result, each agent trained on a total of only three environments (out of eight) for ventricle localization. In addition, the agents shared their ERBs with each other at the end of every iteration to facilitate peer-to-peer experience sharing with lifelong learning.

Each of the agents had different training speeds owing to different GPUs used by each agent. Subsequently, we implemented asynchronous learning, meaning when the agent finishes training on a task, it will broadcast its ERBs to the hub and begin training on the next dataset using its previous ERB and any new ERBs available at the hub. As a result, the number of ERBs available from other ADFLL agents when starting a new round of training will significantly vary between the slowest and the fastest agents in the network. This process is continued until all four agents complete three rounds of training. For comparison, Agent M was trained for eight rounds, and Agent X was trained to use the central aggregation of all eight datasets.

**Cross-Modality 3D Spleen Localization:** While the first experiment evaluated the ability of the ADFLL framework to perform asynchronous experience sharing with lifelong learning, the second experiment evaluated the capability of the ADFLL framework to learn from ERBs that encoded different imaging modalities.

We initialized this experiment with five agents and five training sets, each consisting of 10 volumes from different imaging sequences (PET, CT, MRI-T1, MRI-DIXON-F, MRI-DIXON-W) for the task of 3D spleen localization. At the beginning of each iteration, each agent encounters one of the five datasets to train a deep reinforcement learning model for localization of spleen on that dataset. Each agent was trained for three rounds i.e., each agent sequentially encountered three datasets (out of five). In addition, the agents shared their ERBs with each other at the end of every iteration to facilitate peer-to-peer experience sharing with lifelong learning. At the end of three rounds, we evaluated each ADFLL node on all five tasks on both the validation and external test sets and compared its performance to Agent X (that had central access to all the datasets), Agent M (that had sequential access to all the datasets), and Agent Y (that had access to only the CT dataset)

**Evaluation Metric:** The performance metric was set as the terminal Euclidean distance in pixels between the agent's prediction and the target landmark. We performed paired t-tests to compare the performance of the ADFLL agents with the traditional lifelong learning framework, all-knowing deep reinforcement learning agent, and partial-knowing deep reinforcement learning agent. The p-value for statistical significance was set to $p \leq 0.05$.

## 4. Results

**Cross-Environment 3D Ventricle Localization:** As shown in Table 1, all four ADFLL agents had excellent performance with no significant difference in performance from the all-knowing Agent X ($p > 0.05$). In addition, three agents (A1, A3, and A4) had no significant difference from Agent M (best-case traditional LL agent), and A2 was significantly better than Agent M ($p = 0.01$), just after three rounds of training, compared to eight rounds of training for Agent M. After three rounds of training, A2 was able to achieve a mean

Table 1: Comparison of distance error between ADFLL agents (Agent 1-4) after round 3, all-knowing best case DRL agent (Agent X) after round 1, partially-knowing worst case DRL agent (Agent Y) after round 2.5, and traditional lifelong baseline DRL agent (Agent M) after round 8 on BrATS dataset

| Patient Characteristics | AgentX
Best case | AgentY
Worst case | AgentM
Traditional LL | Agent1 | Agent2
ADFLL Agents | Agent3 | Agent4 |
|---|---|---|---|---|---|---|---|
| Coronal_LGG_t1 | 10.05 | 8.94 | 10.68 | 8.06 | 8.06 | 10.49 | **6.08** |
| Coronal_LGG_t2 | 9.22 | 8.31 | 8.94 | 9.49 | **7.35** | 8.25 | 33.79 |
| Sagittal_LGG_flair | 10.77 | 60.42 | 14.59 | 11.75 | **8.12** | 8.31 | 10.49 |
| Axial_LGG_tice | 7.07 | 89.91 | 16.67 | 31.19 | 6.16 | 6.4 | **3.74** |
| Axial_HGG_flair | 4.47 | 90.05 | 22.16 | 66.56 | **4.24** | 12.04 | 22.09 |
| Sagittal_HGG_t1 | 31.19 | 65.49 | 13.15 | 10.05 | **6.71** | 11.75 | 7.55 |
| Sagittal_HGG_t2 | **10.25** | 68.61 | 11.58 | 24.7 | 12.33 | 22.67 | 12.37 |
| Coronal_HGG_tice | 11.22 | 44.11 | 23.58 | 40.47 | **9.54** | 39.71 | 13.19 |
| Mean | 11.78 | 54.48 | 15.17 | 25.28 | **7.81** | 14.95 | 13.66 |
| Std. dev | 8.16 | 32.05 | 5.32 | 20.46 | **2.4** | 11.17 | 9.87 |
| Ttest (vs. Agent X) | | 0.01 | 0.4 | 0.18 | 0.22 | 0.54 | 0.73 |
| Ttest (vs. Agent M) | 0.4 | 0.01 | | 0.12 | 0.01 | 0.95 | 0.72 |
| Ttest (vs. Agent Y) | 0.01 | | 0.01 | 0.01 | 0 | 0.01 | 0.02 |

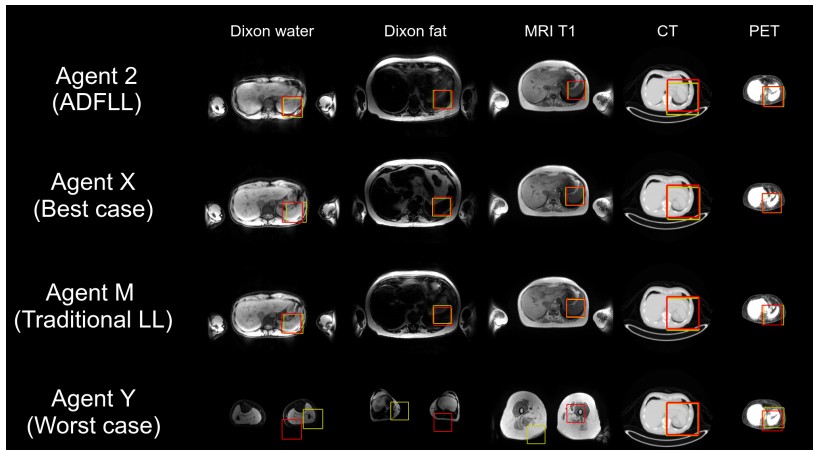

Figure 3: Illustration of the performance of ADFLL agent 2, all-knowing best case DRL agent (Agent X), partially-knowing worst case DRL agent (Agent Y), and and traditional lifelong baseline DRL agent (Agent M) on in-distribution Whole body datasets

distance error of $7.81 \pm 2.4$ on all eight tasks, compared to the $11.78 \pm 8.16$ (p=0.22) for Agent X, significantly lower compared to the lower bound conventional DRL agent, Agent Y ($54.58 \pm 32.05$; $p < 0.001$), and Agent M ($15.17 \pm 5.32$; $p = 0.01$) after eight rounds of training.

**Cross-Modality 3D Spleen Localization:** Table 2 summarizes the performance of all five ADFLL agents along with the three baseline agents, Agent X (all-knowing best-case

DRL agent), Agent M (traditional lifelong learning agent), and Agent Y (DRL agent trained on just the CT dataset) for the internal validation dataset. Agent 2 was the best performing ADFLL agent with the Euclidean distance error of $8.01 \pm 5.47$. As shown in Appendix Table 2, there was no significant difference in the Euclidean distance error between the four (out of five) ADFLL agents and the Agent X ($d = 8.01 \pm 5.47$) for the internal validation dataset. Furthermore, all five agents had no significant difference in Euclidean distance error from the Agent M ($d = 11.22 \pm 20.09$). Furthermore, all five ADFLL agents had a significantly lower Euclidean distance error compared to Agent Y ($d = 117.82 \pm 70.31$) that was trained only the CT dataset. Figure 3 illustrates the performance of the best performing ADFLL agent, Agent 2 as well as the baseline agents across all five imaging sequences.

Appendix Tables 3, 4, and 5 summarize the performance of all ADFLL and baseline agents across the three external datasets. Apart from spleen MSD CT dataset where Agent M significantly ($p < 0.05$) outperformed three (out of five) ADFLL agents, there was no significant difference between the ADFLL agents and the best-case baseline agents, X and M across all the experiments.

## 5. Discussion

In this work, we proposed the asynchronous decentralized federated learning framework to achieve collective intelligence via continual learning from peers with excellent results across all experiments. The ADFLL framework's performance was similar to upper bound baselines: the all-knowing baseline with all the datasets centrally aggregated in one location and the conventional LL baseline, where an agent sequentially trains across all the datasets.

The cross-environment ventricle localization experiment allowed us to evaluate the real-world setting where different agents with different compute environments train at different speeds leading to asynchronous communication. Despite training asynchronously and on different sets of shared experiences, all four agents had a similar performance to the conventional lifelong learning model. However, the fastest agent (A1) had the worst performance across all ADFLL agents owing to being trained on only a subset of experiences shared across the network. Similarly, the cross-modality spleen localization experiment allowed us to evaluate the capability of ADFLL agents to learn from peer ERBs encoding different modalities from disparate datasets. Our results that ADFLL agents are not only capable of learning from ERBs that encode significantly different imaging environments, but also demonstrate excellent generalization performance when when evaluated on external datasets with different modalities.

Asynchronous federated learning has also been explored in other areas (Chen et al., 2019). They offer the ability to deal with nodes with different computational power but lack the decentralization that allows the system to be more flexible. Similarly, in (Liu et al., 2022; Huang et al., 2022), the cost of removing a central node is a quadratic complexity communication scheme in that every node communicates with every node. Our work has certain limitations. This preliminary study focused on a single landmark localization task with deep reinforcement learning. Our future work will expand this framework to multi-agent systems and other medical imaging tasks such as segmentation and classification. In conclusion, we demonstrated a privacy-aware, asynchronous, decentralized federated learning system with robust and efficient system topology.

## Acknowledgments

This work was supported by the DARPA grant: DARPA-PA-20-02-11-HR00112190130 and 5P30CA006973 (Imaging Response Assessment Team-IRAT), U01CA140204. The MRI test data used in this publication were generated by the National Cancer Institute's Cancer Moonshot Biobank and National Cancer Institute Clinical Proteomic Tumor Analysis Consortium (CPTAC).

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

## Appendix A. Clinical Data

Figure 4 illustrates the eight environments used for the cross-environment 3D ventricle localization experiment.

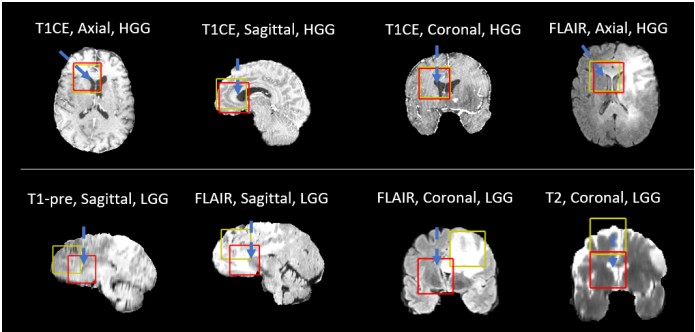

Figure 4: Illustration of the 8 task-environment pairs. The red boxes indicate the true landmark location of the top left ventricle. The yellow box is a predicted location from ADFLL agents during their training progressions

## Appendix B. Simulation Experiments

We conducted multiple simulation experiments to evaluate our framework's scalability, flexibility, and robustness. We evaluated the agents' average performance to localize the top left ventricle across all 24 imaging environments for both experiments. Additionally, since it is prohibitively expensive to experiment on 24 different machines, these systems were simulated on the NVIDIA DGX-1 with a synchronous training protocol.

### B.1. Addition and Deletion of agents:

**Addition experiment:** We initialized this experiment with a system of four ADFLL agents. We subsequently increased the number of agents in the system from 4 to 16 over 4 rounds (4,8,12,16). At the beginning of every round, each ADFLL agent in the system receives a new training dataset with a different imaging environment. Each agent performs ADFLL using ERBs from its previous tasks in addition to ERBs being communicated across the network from other agents. We further simulated a communication dropout of 75% to account for network communication issues in the real world, leading to information loss while transmitting ERBs across agents. This experiment aimed to demonstrate how newer agents joining the system at different points in time can take advantage of the information within the system and learn the collective knowledge available in the system within just one round.

   **Deletion experiment:** In the deletion experiment, we gradually decreased The number of agents in the system from 24 to 1 agent over the progression of 5 rounds (24,12,6,3,1). Similar to the addition experiment, each agent remaining in the system receives a new training dataset every round, and the ERBs are communicated across the network. The communication for this experiment was also simulated with a 75% dropout. This experiment aimed to demonstrate how the proposed ADFLL framework preserves the collective knowledge in a lifelong learning manner across all the tasks, even as the agents contributing the knowledge leave the system.

*Results:* In the two simulation studies, our framework showed scalability of up to 24 agents, robustness against network dropout, and flexibility in system topology. As shown in Fig. 5, we see that the average Euclidean distance error across all agents and all 24 tasks decreases as more agents are added to the system, with an average Euclidean distance error of $16.89 \pm 16.34$ at the end of 4 rounds. This demonstrates the capability of ADFLL agents to directly train from peer experiences without pre-training, allowing newer agents joining the system directly learn existing knowledge within the system in one round. For the deletion experiment, the average Euclidean distance error across all agents decreases while half of the agents are deleted every round, resulting in an average Euclidean distance error of $8.55 \pm 7.12$ after the final remaining agent at the end of 5 rounds (shown in Fig.5). In comparison, the average Euclidean distance error was $8.34 \pm 7.26$ ($p > 0.05$) for Agent X and $8.15 \pm 5.42$ ($p > 0.05$) for Agent M. This shows that the knowledge agents learned and captured in ERBs are not lost when agents are removed from the system. When agents are added, the new agents can catch up with existing agents in one round. Moreover, the 75% dropout rate applied to every round of both experiments shows the robustness of our framework against network failures, a significant bottleneck for federated learning frameworks.

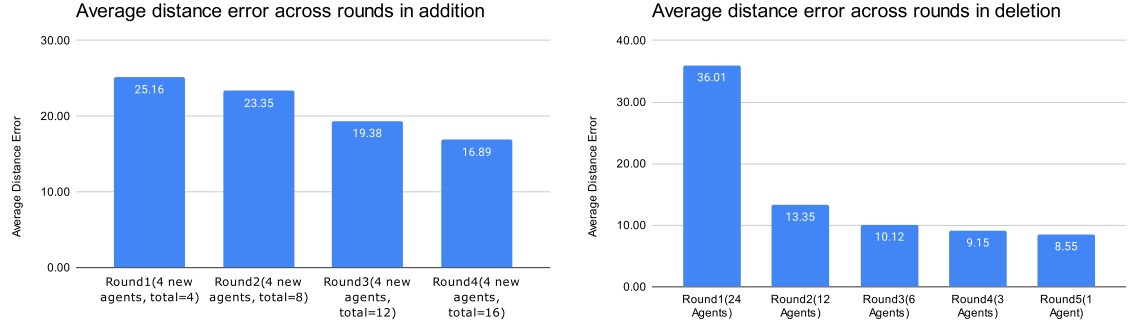

Figure 5: Left: Comparison of distance error of all agents in the system across 4 rounds of training as agents join the system. Right: Comparison of distance error of all agents in the system across 4 rounds of training as agents leave the system.

## B.2. Scalability Experiments

### B.2.1. SYSTEMS OF 2,4,8,16,24 AGENTS

To test the influence of agent count in a system on its performance, we tested systems with 2, 4, 8, 16, or 24 Agents training for two rounds on their performance to localize the top left ventricle in all 24 imaging environments. Since it is prohibitively expensive to experiment on 24 different machines, these systems were simulated on the NVIDIA DGX-1 with a synchronous training protocol.

*Result:* Figure 6 demonstrates the comparison of Agent Rewards (AR) when different ADFLL systems consisting of 2, 4, 8, 16, and 24 agents were trained for two rounds in addition to the best outcome agent. The ADFLL agent systems with $\geq 4$ agents learned

more than 80% of the environments within just two rounds. Finally, the ADFLL systems with 16 and 24 agents learned greater than 90% of the environments.

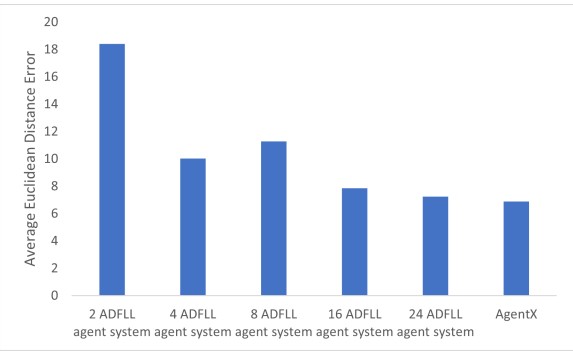

Figure 6: Comparison of Euclidean distance error between different experimental setups involving all the base cases and ADFLL setups with 2, 4, 8, 16, and 24 agents

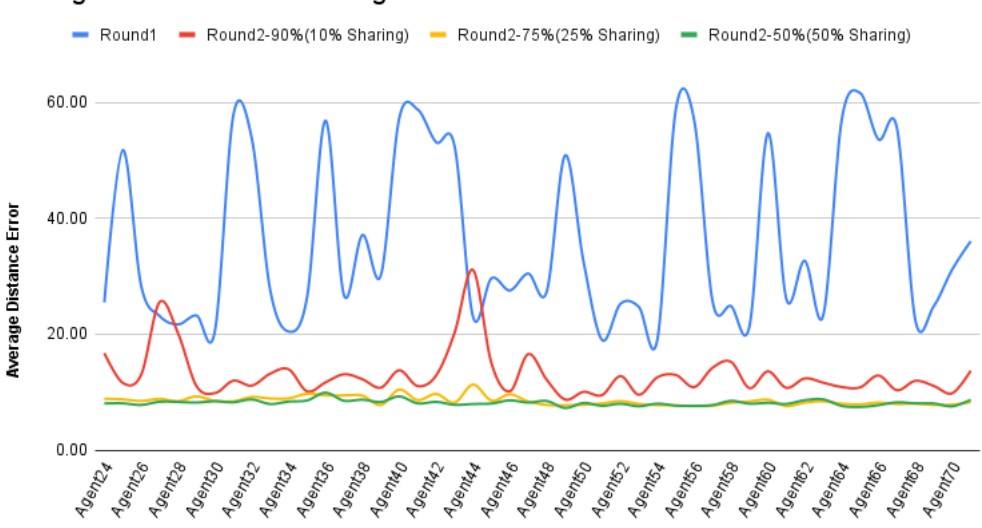

Figure 7: Illustration of the average distance error for the 48 new agents using different dropout rates

### B.3. 72 Agents with significant network dropout

To further test the scalability limit of our framework, we designed an experiment that involves 72 agents. We will use the 24 agents from the previous experiments and add 48 agents to the system, and the 48 agents will run for two rounds. They will be randomly

assigned to an imaging environment each round. In the first round, they will learn from their local data and ERBs of the previous 24 agents. In the second round, they will learn from their local data, ERBs of the previous 24 agents, and ERBs from the first round of the 48 agents. Additionally, the robustness of our framework against communication failure is tested, and $50\%, 75\%, 90\%$ dropout rate is applied to the number of ERBs each agent receives, meaning only $50\%, 25\%, 10\%$ ERBs reached the agents.

*Result:* Figure 7 shows the difference in average distance error when using different dropout rates during ERB sharing. $90\%$ dropout rate performs the worst out of the three dropout rates, with higher average distance error and higher deviation in performance. Performance $75\%$ and $50\%$ are very similar, with $50\%$ dropout rates having a slightly lower average distance error. Compared to no sharing in round 1, sharing just a tiny fraction ($10\%$) of the ERB in the system still can have a massive boost in performance. Figure 7 shows the highest dropout rate the new 48 agents can accommodate without losing significant performance.

## Appendix C. Internal and External Validation results for cross-modality 3D spleen localization

Table 2: Comparison of distance error between ADFLL agents (Agent 0-4) after round 3, all-knowing best case DRL agent (Agent X) after round 1, partially-knowing worst case DRL agent (Agent Y) after round 1, and traditional lifelong baseline DRL agent (Agent M) after round 5 on the in-distribution whole-body dataset.

| Image source | AgentX | AgentM | AgentY | agent0 | agent1 | agent2 | agent3 | agent4 |
|---|---|---|---|---|---|---|---|---|
| Mean | 8.01 | 11.32 | 117.82 | 9.77 | 9.87 | **8.01** | 10.55 | 11.33 |
| Std. dev | 5.47 | 20.09 | 70.31 | 9.67 | 7.59 | **5.47** | 13.05 | 16.24 |
| TTEST with X | | 0.34 | 0.00 | 0.20 | 0.02 | 1.00 | 0.16 | 0.22 |
| TTEST with M | 0.34 | | 0.00 | 0.68 | 0.68 | 0.33 | 0.84 | 1.00 |
| TTEST with Y | 0.00 | 0.00 | | 0.00 | 0.00 | 0.00 | 0.00 | 0.00 |

Table 3: Comparison of distance error between ADFLL agents (Agent 0-4) after round 3, all-knowing best case DRL agent (Agent X) after round 1, partially-knowing worst case DRL agent (Agent Y) after round 1, and traditional lifelong baseline DRL agent (Agent M) after round 5 on the out-of-distribution MSD CT dataset.

| Image source | agentX | agentM | agentY | agent0 | agent1 | agent2 | agent3 | agent4 |
|---|---|---|---|---|---|---|---|---|
| Mean | 10.65 | **8.87** | 13.78 | 9.31 | 11.17 | 11.24 | 10.78 | 10.72 |
| Std. dev | 4.88 | **3.68** | 11.28 | 4.51 | 6.57 | 4.46 | 5.17 | 6.13 |
| TTEST(vs AgentX) | | 0.01 | 0.09 | 0.11 | 0.52 | 0.24 | 0.83 | 0.94 |
| TTEST(vs AgentM) | 0.02 | | 0.01 | 0.70 | 0.04 | 0.00 | 0.01 | 0.08 |
| TTEST(vs AgentY) | 0.07 | 0.00 | | 0.01 | 0.21 | 0.19 | 0.09 | 0.08 |

Table 4: Comparison of distance error between ADFLL agents (Agent 0-4) after round 3, all-knowing best case DRL agent (Agent X) after round 1, partially-knowing worst case DRL agent (Agent Y) after round 1, and traditional lifelong baseline DRL agent (Agent M) after round 5 on the out-of-distribution TCIA MRI dataset.

| Image Source | agentX | agentM | agentY | agent0 | agent1 | agent2 | agent3 | agent4 |
|---|---|---|---|---|---|---|---|---|
| Mean | 30.98 | 19.05 | 133.75 | **18.00** | 23.00 | 25.83 | 21.30 | 27.48 |
| Std. dev | 34.00 | 16.48 | 73.81 | **10.47** | 21.68 | 37.25 | 23.63 | 17.73 |
| TTEST(vs AgentX) | | 0.33 | 0.01 | 0.29 | 0.53 | 0.77 | 0.46 | 0.78 |
| TTEST(vs AgentM) | 0.33 | | 0.00 | 0.82 | 0.49 | 0.62 | 0.50 | 0.24 |
| TTEST(vs AgentY) | 0.01 | 0.00 | | 0.00 | 0.00 | 0.00 | 0.00 | 0.00 |

Table 5: Comparison of distance error between ADFLL agents (Agent 1-4) after round 3, all-knowing best case DRL agent (Agent X) after round 1, partially-knowing worst case DRL agent (Agent Y) after round 1, and traditional lifelong baseline DRL agent (Agent M) after round 5 on the out-of-distribution TCIA PET dataset.

| Image source | agentX | agentM | agentY | agent0 | agent1 | agent2 | agent3 | agent4 |
|---|---|---|---|---|---|---|---|---|
| Mean | 20.21 | 20.37 | 105.78 | 18.92 | 20.77 | **18.04** | 22.62 | 18.62 |
| Std. dev | 30.35 | 30.52 | 64.22 | 30.15 | 30.33 | 30.50 | **29.67** | 29.83 |
| TTEST(vs AgentX) | | 0.93 | 0.01 | 0.25 | 0.72 | 0.24 | 0.40 | 0.32 |
| TTEST(vs AgentM) | 0.93 | | 0.01 | 0.46 | 0.77 | 0.34 | 0.44 | 0.28 |
| TTEST(vs AgentY) | 0.01 | 0.01 | | 0.01 | 0.01 | 0.01 | 0.01 | 0.01 |

