# OpenReview forum: "Towards a Collective Medical Imaging AI: Enabling Continual Learning from Peers"
_MIDL.io/2024/Conference — MIDL 2024 Poster_

### Official Review · Reviewer_o3pk · 2024-02-27

**Confidence:** 4
**Preliminary Rating:** 2
**Final Rating:** 3.5

**Summary:**

An asynchronous decentralized federated lifelong learning (ADFLL) method that allows lifelong learning agents to continually learn from their own previous experiences, as well as experiences from other nodes in the network, was proposed, thus overcoming the potential drawbacks of conventional federated learning. An evaluation of the ADFLL framework in two experimental setups was setup: a simulation setup where we simulate addition and deletion of nodes in a FL setup and a deployment setup where different nodes working on different tasks are initialized on different machines with different compute.

**Strengths:**

1. The authors introduced a new framework that combines federated lifelong learning with deep reinforcement learning, which enables asynchronous training schedules and allows learning agents to continually learn from their own experience.
2. The experiment is sufficient in BraTS dataset.

**Weaknesses:**

1. The authors claimed that they developed a new framework, but they failed to explain the details.
2. The authors failed to explain the combination of federated lifelong learning and deep reinforcement learning.
3. Experiments are not sufficient. Only one dataset was included, and no baseline comparison.

**Detailed Comments:**

The reviewer would like to take this paper as a validation paper if the experiments are sufficient.

**Justification Of Final Rating:**

The authors lay the foundation of a promising federated learning system that allows for agents to learn from the experience of other agents. They found no significant differences between their system and the baseline model that had access to all training data highlighting ADFLL's effectiveness.

The reviewer would like to recommend to accept this article, I cannot be confident on this as I am not an expert in Reinforcement Learning.

**Justification Of The Preliminary Rating:**

The paper isn't good enough in its current form, the authors failed to explain either the method or the experiments in detail. The authors should take into consideration that this paper should be presented as a validation paper. The reviewer would like to raise the score if the authors like to supplement the experiment or convince me that the experiments are sufficient.

**Questions To Address In The Rebuttal:**

Please see the weaknesses.
The authors should address the novelty of their method in rebuttal.

---

> ### Author Response · Authors · 2024-03-18
> **Rebuttal to Reviewer 5**
>
> Thank you for taking the time to provide valuable insights on our submission. Please see below the point-by-point response to the comments (in bold):
>
> **1. The authors claimed that they developed a new framework, but they failed to explain the details. The authors failed to explain the combination of federated lifelong learning and deep reinforcement learning.**
>
> We have updated our introduction and methods sections to improve readability and explain the combination of federated lifelong learning and deep reinforcement learning. In addition, we have also added a new Figure 1 to illustrate the combination between peer-to-peer experience sharing, lifelong learning, and deep reinforcement learning.
>
> We have added the following paragraph to the introduction
> "We develop and evaluate ADFLL using deep reinforcement learning (DRL) as the base model architecture at each node for the task of 3D landmark localization in medical imaging volumes. DRL with LL capability using selective experience replay (SERIL) has previously been validated for continual landmark localization across different imaging environments using DRL for medical imaging and also forms the basis of LL for each node in ADFLL"
>
>
>
>
> **2. Experiments are not sufficient. Only one dataset was included.**
>
> We have added a new experiment to address this limitation of our original submission. The new experiment focuses on cross-domain 3D localization of spleen on five different imaging sequences from three modalities: MRI Dixon water, MRI Dixon fat, MRI T1, CT, and PET. In addition to learning from ERBs encoding different modalities, we also evaluated the trained ADFLL agents on multiple external datasets from Medical Segmentation Decathlon (MSD) and The Cancer Imaging Archive (TCIA). **The clinical data and the experimental details are provided in Sections 3.2 and 3.3 in the updated manuscript**
>
> **3. No baseline comparison**
>
> The ADFLL framework proposed in this work evaluates the capability of lifelong learning agents to learn from each others' experiences for cross-domain landmark localization in medical images. Across all our experiments, we have included comparison to two upper-bound and one lower-bound baseline models, detailed below:
>
> Agent X: The "upper bound" best case scenario where all the tasks being shared across the network are centrally available for training.
>
> Agent Y: The "lower bound" worst case scenario for centralized training, when the agent is only presented with 1 task.
>
> Agent M: The "upper bound" lifelong learning scenario for centralized training, when tasks and data are available to the agent sequentially (one dataset available after each training round). This provides comparison between training from ones own experiences (traditional LL) and training from each others experiences (ADFLL proposed in this work)

---

### Official Review · Reviewer_yE8T · 2024-02-28

**Confidence:** 3
**Preliminary Rating:** 3
**Final Rating:** 4

**Summary:**

The paper presents the asynchronous decentralized federated lifelong learning (ADFLL) method as a solution to the limitations of traditional federated learning, enabling lifelong learning in a decentralized, privacy-preserving manner. ADFLL allows agents to continuously learn from both personal and network-wide experiences, efficiently integrating new tasks and nodes. Experiments use the BRATS dataset in simulation and deployment setups, and ADFLL demonstrated remarkable performance in retaining knowledge and adapting to new agents, matching or outperforming centralized and conventional lifelong learning approaches.

**Strengths:**

- Integrating federated learning with lifelong learning in an asynchronous and decentralized manner is a promising and significant topic in the field of federated learning.
- The experimental evaluation is comprehensive, especially the evaluations on the impact of the number of agents in a system on its performance.

**Weaknesses:**

- The motivation behind the study is somewhat unclear; the authors should more explicitly emphasize the significance of the ADFLL framework for medical imaging tasks, particularly outlining the primary motivation for evaluating landmark localization tasks using the BraTS dataset.
- Additionally, the rationale for incorporating deep reinforcement learning within the ADFLL framework is not immediately apparent. From my perspective, this task could potentially be accomplished through a straightforward supervised learning approach.

**Detailed Comments:**

N/A

**Justification Of Final Rating:**

I am satisfied with the author's response regarding the motivation and the integration of reinforcement learning, addressing my concerns. And I accordingly increase my final score. Additionally, it would be beneficial to provide further clarification on the IID Federated Learning (FL) setting and explain why the current version focuses exclusively on the IID setting.

**Justification Of The Preliminary Rating:**

My primary concerns relate to the motivation and the importance of the proposed framework within the field of federated learning with medical imaging. Additionally, a detailed explanation regarding the integration of deep reinforcement learning within the ADFLL framework is necessary.

**Questions To Address In The Rebuttal:**

In the description of section 2.5, the client data partitioning appears to be divided into IID distributions, failing to introduce substantial data heterogeneity across different clients.

---

> ### Author Response · Authors · 2024-03-18
> **Rebuttal to Reviewer 4**
>
> Thank you for taking the time to provide valuable insights on our submission. Please see below the point-by-point response to the comments (in bold):
>
> **1. The motivation behind the study is somewhat unclear; the authors should more explicitly emphasize the significance of the ADFLL framework for medical imaging tasks, particularly outlining the primary motivation for evaluating landmark localization tasks using the BraTS dataset.**
>
> Thank you for your feedback. we have clarified and emphasized the significance of our framework for medical imaging tasks in the abstract and introduction. More specifically, we emphasized the capability of our framework to learn from cross-domain experiences shared by different agents in the network enabling learning in newer unseen domains. To further emphasize the significance of the proposed framework, we have added a new experiment focused on cross-domain 3D localization of spleen on five different imaging sequences from three modalities: MRI Dixon water, MRI Dixon fat, MRI T1, CT, and PET. In addition to learning from ERBs encoding different modalities, we also evaluated the trained ADFLL agents on multiple external datasets from Medical Segmentation Decathlon (MSD) and The Cancer Imaging Archive (TCIA). **The clinical data and the experimental details are provided in Sections 3.2 and 3.3 in the updated manuscript**
>
> **2. Additionally, the rationale for incorporating deep reinforcement learning within the ADFLL framework is not immediately apparent. From my perspective, this task could potentially be accomplished through a straightforward supervised learning approach.**
>
> The proposed ADFLL framework is compatible with any memory based deep lifelong model as the base architecture at different nodes. This work uses deep reinforcement learning (DRL) as an example to demonstrate and evaluate the capabilities of the ADFLL framework. While the task of 2D object detection can be accomplished using a straightforward object detector, DRL has demonstrated excellent performance in tasks involving 3D landmark localization in the literature [1, 2]. Therefore, we evaluated the ADFLL framework for 3D landmark localization in this work.
>
> **3. In the description of section 2.5, the client data partitioning appears to be divided into IID distributions, failing to introduce substantial data heterogeneity across different clients.**
>
> The reviewer raises an important point. We agree that our initial experiment sampled all the environments from the same imaging dataset leading to IID distributions. Therefore, we added a new experiment focused on cross-domain 3D spleen localization to demonstrate the capability of the proposed ADFLL framework in non-iid settings. Please refer to our response #1. The new experiment samples environments from three different modalities from two distinct datasets acquired using different scanners (PET/CT vs. MRI). Furthermore, we evaluated our models on multiple external datasets sourced from MSD and TCIA.
>
> References
>
> 1. Florin-Cristian Ghesu, Bogdan Georgescu, Yefeng Zheng, Sasa Grbic, Andreas Maier, Joachim Hornegger, and Dorin Comaniciu. Multi-scale deep reinforcement learning for real-time 3d-landmark detection in ct scans. IEEE transactions on pattern analysis and
> machine intelligence, 41(1):176–189, 2017.
> 2. Amir Alansary, Ozan Oktay, Yuanwei Li, Loic Le Folgoc, Benjamin Hou, Ghislain Vaillant, Konstantinos Kamnitsas, Athanasios Vlontzos, Ben Glocker, Bernhard Kainz, and Daniel Rueckert. Evaluating Reinforcement Learning Agents for Anatomical Landmark
> Detection. Medical Image Analysis, 2019.

---

### Official Review · Reviewer_kUJE · 2024-03-03

**Confidence:** 5
**Preliminary Rating:** 2
**Final Rating:** 4

**Summary:**

The paper introduces an Asynchronous Decentralized Federated Lifelong Learning (ADFLL) framework aimed at overcoming the limitations of conventional federated learning in medical imaging, specifically landmark localization. It allows for continual learning from peers without a central node, enabling the system to adapt to new tasks and environments without catastrophic forgetting. The framework was evaluated using the brain tumor segmentation (BraTS) dataset, demonstrating its efficacy in maintaining high performance across various imaging environments and under conditions of significant agent turnover and network dropout. The ADFLL framework showed robustness, scalability, and flexibility, with performances comparable to centralized baselines, making it a promising approach for real-world applications in medical imaging AI.

**Strengths:**

The strength of this paper lies in its innovative approach to federated learning in medical imaging. It introduces a novel Asynchronous Decentralized Federated Lifelong Learning (ADFLL) framework that addresses the challenges of scalability, robustness, and continuous learning in dynamic and decentralized environments. The paper demonstrates the framework's effectiveness in maintaining high performance across varied imaging tasks without the need for a central coordinating node, showcasing its potential for real-world applications where data privacy and system adaptability are critical.

**Weaknesses:**

1. For section 3.3, can you break into sections to clearly explain your experiments? This is the most important part of your article, it is very confusing to read. I spent a lot of time reading this section and still diffucult to read and understand.
a) "As shown in Fig.3, we also see that the average Euclidean distance error across all agents decreases, while half of the agents are deleted every round, resulting in an average Euclidean distance error of 8.55 ± 7.12 after for the final remaining agent at the end of 5 rounds."
This sentence is difficult to understand, why error decreases?
b) also, you need to make your table or your content explicitly clear, what are X, Y, M, which one is the upper and lower bound?  We don't remember, we need to go back to your content to search again.

2. Because there is a page limit for htis submission, you need to reduce the length of section 3.2, section 4 as well as section 1. You need to use more paragraphs to clearly explain your work to authors
3. The plot (Figure 3) and table 1 cannot be screenshots. They are not acceptable. We may question about the authenity of the experiment.

**Detailed Comments:**

1. The abstract is too long
2. Figure 1 is good, however some fonts are too small
3. page 4, last line, "upper bound" quotation mark, similiarly, page 5 "lower bound"
4. page 5, "ventricl", typo
5. Table 1, using a screenshot to show the result is not acceptable. Also, some key results need to be high lighted.
6. Table 1 results are not clearly explained, it is not striaght forward to read and understand the result

**Justification Of Final Rating:**

Based on the author's rebuttal and updated paper, this paper has improved, and the author successfully addressed some problems. I think this paper has shown a good work.
If this paper can be accepted, I will suggest the author could improve the writing of this paper, to help people to understand easier.

**Justification Of The Preliminary Rating:**

This work is novel and has very interesting ideas. However, the poor writing styles and some unclear contents made people hard to read and understand. Especially section 3.3, it seems there is a lot of work, however, when I read this section, i think the contribution is minor. Therefore this paper cannot have a high score.

**Questions To Address In The Rebuttal:**

Can you re-explain your experiment, i.e. section 3.3 again?

---

> ### Author Response · Authors · 2024-03-18
> **Rebuttal to Reviewer 3**
>
> Thank you for taking the time to provide valuable insights on our submission.
>
> - We have shortened and clarified the abstract
> - We have updated Figure 1 (now is Figure 2) so the text is more readable.
> - We have thoroughly revised and checked the typos in our paper.
> - We have updated experimental details for all our experiments to improve readability.
> - We have updated the table of results and highlighted key results.
> - We have updated the results section to explain the table of results better.
>
> We have added an additional experiment to our manuscript to highlight our contribution. The new experiment focuses on cross-domain 3D organ localization of the spleen on five different imaging sequences from three modalities: MRI Dixon water, MRI Dixon fat, MRI T1, CT, and PET. We have also validated our model performance on external datasets: the MSD spleen dataset, PET and MRI datasets from The Cancer Imaging Archive. **The clinical data and the experimental details are provided in Sections 3.2 and 3.3 in the updated manuscript**
>
> We hope this addresses all the concerns you have with our paper. Again, we would like to thank you for your time and feedback!

---

### Official Review · Reviewer_5h7P · 2024-03-03

**Confidence:** 3
**Preliminary Rating:** 2

**Summary:**

This work proposes Asynchronous Decentralized Federated Lifelong Learning (ADFLL). This method combines the advantages of federated learning and lifelong learning, enabling multiple agents to collaboratively train models without sharing raw data. Compared to traditional federated learning, the ADFLL method does not require a central node, can handle data and agent heterogeneity, and can continuously learn multiple tasks, avoiding catastrophic forgetting.

Deep Reinforcement Learning (DRL) framework: The paper proposes a DRL framework that utilizes the Deep Q-Network (DQN) algorithm for landmark localization tasks. This framework demonstrates flexibility and effectiveness in handling medical imaging data and can adapt to different task environments.

Lifelong learning with selective experience replay: To prevent catastrophic forgetting, the paper employs selective experience replay, focusing on selected experiences from past tasks. This approach enables the model to retain knowledge of previous tasks while learning new ones, thereby enhancing generalization and accuracy.

Distributed database system: To manage communication among agents effectively, the paper adopts a distributed database system that connects agents with central nodes, facilitating asynchronous communication and experience sharing. This system architecture ensures robustness in the event of node or central node failures and manages communication complexity efficiently.

In summary, the paper introduces a novel approach that integrates federated learning, lifelong learning, and deep reinforcement learning, effectively addressing privacy, data heterogeneity, and catastrophic forgetting issues in medical image analysis, thereby offering new solutions to the field.

**Strengths:**

The advantages of the paper are as followings:
The paper comes up with fresh ways to use computers to look at medical images, making it easier to keep patient info private and learn from different types of data.
It's like having a tool that can handle different jobs without breaking a sweat. This method can deal with lots of different medical image tasks without a hitch.
The paper don't just talk about fancy theories. They show you step-by-step how to use their ideas in the real world.

**Weaknesses:**

While the method performs well in medical image analysis, it seems to be a general approach rather than specifically designed for medical imaging. This may suggest that it lacks optimization for specific medical field requirements, potentially resulting in suboptimal performance in certain scenarios.

The figures and charts in the paper may lack aesthetic appeal, which could affect readers' comprehension and engagement with the content. Visually appealing illustrations and charts can better showcase methods and results, enhancing the paper's readability and attractiveness. (core drawback)

The paper may lack detailed comparative experiments with other similar methods, making it difficult for readers to assess the method's performance advantages. Comparative experiments can help validate the method's superiority and its applicability in different scenarios.  (core drawback)

The method may involve complex algorithms and computational processes, potentially leading to high computational costs in practical applications or requiring more computational resources support. (small drawback)

If the models or algorithms involved in the method lack sufficient interpretability, it may face challenges in explaining results and credibility in practical applications, limiting its acceptance and application range in clinical settings.  (small drawback)

**Detailed Comments:**

The drawbacks that need to be addressed are all listed above.

**Justification Of The Preliminary Rating:**

While the method shows promise in analyzing medical images, it appears to be a general solution rather than specifically tailored for medical imaging needs. This lack of specialization could lead to suboptimal performance in certain situations where specific medical requirements are crucial. Moreover, the paper's visuals, such as figures and charts, lack aesthetic appeal, potentially hindering readers' comprehension and engagement with the content. Improving the visual presentation could enhance the paper's readability and attractiveness. Additionally, the absence of detailed comparisons with similar methods makes it challenging for readers to gauge the method's performance advantages. Comparative experiments are vital for validating its superiority and applicability across various scenarios. Furthermore, the method's complexity and computational demands may pose practical challenges, potentially requiring significant computational resources or leading to high costs. Lastly, if the models or algorithms lack interpretability, it could hinder their acceptance and usability in clinical settings, as healthcare professionals may struggle to understand and trust the results.

**Questions To Address In The Rebuttal:**

I recommend authors not to rebuttal, they should consider submitting the paper after improving the effectiveness of experimental analysis and enhancing the aesthetics and visual appeal of the figures and charts.

---

> ### Author Response · Authors · 2024-03-18
> **Rebuttal to Reviewer 2**
>
> Thank you for taking the time to provide valuable insights on our submission. Please see below the point-by-point response to the comments (in bold):
>
> **While the method performs well in medical image analysis, it seems to be a general approach rather than specifically designed for medical imaging. This may suggest that it lacks optimization for specific medical field requirements, potentially resulting in suboptimal performance in certain scenarios.**
>
> We agree with the reviewer that our framework is a general approach. However, while this study can be generally applied to any application, we have targeted the specific problem of 3D landmark localization for medical imaging. We have expanded our experiments to include 3D cross-modality organ localization across different modalities (PET, CT, and MRI) to demonstrate the capability of the proposed method in medical imaging. Furthermore, we have included validation on external out-of-distribution datasets as well. **The clinical data and the experimental details are provided in Sections 3.2 and 3.3 in the updated manuscript**
>
>
> **The figures and charts in the paper may lack aesthetic appeal, which could affect readers' comprehension and engagement with the content. Visually appealing illustrations and charts can better showcase methods and results, enhancing the paper's readability and attractiveness. (core drawback)**
>
> We thank the reviewer for giving us feedback on the representation of our framework using figures and charts. We have improved them and the overall clarity of the manuscript to make it more visually appealing and easy to follow.
>
>
> **The paper may lack detailed comparative experiments with other similar methods, making it difficult for readers to assess the method's performance advantages. Comparative experiments can help validate the method's superiority and its applicability in different scenarios. (core drawback)**
>
> The ADFLL framework proposed in this work evaluates the capability of lifelong learning agents to learn from each others' experiences for cross-domain landmark localization in medical images. Across all our experiments, we have included comparison to two upper-bound and one lower-bound baseline models, detailed below:
>
> Agent X: The "upper bound" best case scenario where all the tasks being shared across the network are centrally available for training.
> Agent Y: The "lower bound" worst case scenario for centralized training, when the agent is only presented with 1 task.
> Agent M: The "upper bound" lifelong learning scenario for centralized training, when tasks and data are available to the agent sequentially (one dataset available after each training round). This provides comparison between training from ones own experiences (traditional LL) and training from each others experiences (ADFLL proposed in this work)
>
> **The method may involve complex algorithms and computational processes, potentially leading to high computational costs in practical applications or requiring more computational resources support. (small drawback)**
>
> The reviewer brings up an excellent point about high computational costs in practical applications. However, the primary goal of this work was to demonstrate a proof of concept in peer-to-peer experience sharing with lifelong learning for medical imaging applications. In this work, we did address the issue of high network bandwidth requirement in an all-to-all communication setup with the hub-node-hub optimized communication setup. In addition, different techniques have been proposed in the literature to reduce the size of ERBs [1] as well as reduce the overall computational complexity of lifelong learning through image compression [2]. In our future work, we plan to integrate different optimization approaches for reduce our framework's overall computational complexity.
>
> **If the models or algorithms involved in the method lack sufficient interpretability, it may face challenges in explaining results and credibility in practical applications, limiting its acceptance and application range in clinical settings. (small drawback)**
>
> The reviewer brings up another excellent point regarding algorithmic interpretability. This was also a primary motivation behind using memory based lifelong learning technique in our work. The selective experience replay buffers shared between agents in our framework are transparent and interpretable. In case of failures, ERBs shared between different agents can be evaluated and improved in its information content.
>
> References
>
> [1] Zheng, Guangyao, et al. "Selective experience replay compression using coresets for lifelong deep reinforcement learning in medical imaging." Medical Imaging with Deep Learning. PMLR, 2024.
>
> [2] Zheng, Guangyao, et al. "A framework for dynamically training and adapting deep reinforcement learning models to different, low-compute, and continuously changing radiology deployment environments." arXiv preprint arXiv:2306.05310 (2023).

---

### Official Review · Reviewer_Cnsd · 2024-03-04

**Confidence:** 3
**Preliminary Rating:** 4
**Recommendation:** Oral
**Final Rating:** 4

**Summary:**

The authors propose a privacy-focused, asynchronous federated learning system that allows for efficient addition and removal of agents to and from the system. The approach was evaluated on anatomy localization based on the BraTS dataset.

**Strengths:**

The authors thoroughly evaluate their proposed scenarios, carrying out a deployment experiment as well, uncovering how real-world scenarios might impact the performance of the system. The evaluated agents (X,Y,M  1-4) are well-justified, establishing reasonable baselines.

**Weaknesses:**

The results are preliminary, implementing the system for a task with a larger dataset would better describe the effectiveness of the agents learning from each other. The small number of training iterations might also affect the presented results.

**Detailed Comments:**

- The reviewer believes the manuscript would benefit greatly with a publicly available code repository describing how to set up ADFLL for the interested reader.

**Justification Of Final Rating:**

The authors have appropriately addressed all my concerns, and I would like to thank them for actively participating in the rebuttal. Acknowledging the comments of the other reviewers, and the explanations I got from the authors for my review, I am keeping my original score.

**Justification Of The Preliminary Rating:**

The authors lay the foundation of a promising federated learning system that allows for agents to learn from the experience of other agents. They found no significant differences between their system and the baseline model that had access to all training data highlighting ADFLL's effectiveness.

**Questions To Address In The Rebuttal:**

- Please address how the number of training rounds were determined. Wouldn't it show a more complete picture of the system if all agents were trained until convergence on a validation dataset?
- Could the authors describe more technical details of what experience (eg. its dimensions) is shared between the agents?

---

> ### Author Response · Authors · 2024-03-18
> **Rebuttal to Reviewer 1**
>
> Thank you for taking the time to provide valuable insights on our submission. Please see below the point-by-point response to the comments (in bold):
>
> **The reviewer believes the manuscript would benefit greatly with a publicly available code repository describing how to set up ADFLL for the interested reader.**
>
> Thank you for your feedback. The code is publicly available at https://github.com/vishwaparekh/ADFLL. We have added the link to the manuscript.
>
> **The results are preliminary, implementing the system for a task with a larger dataset would better describe the effectiveness of the agents learning from each other. The small number of training iterations might also affect the presented results.**
>
> We agree with the reviewer that the results are preliminary, and we would like to address this with additional experiments we conducted that involve landmark localization of the spleen on five different imaging sequences from three modalities: MRI Dixon water, MRI Dixon fat, MRI T1, CT, and PET. We have also validated our model performance on external datasets from Medical Segmentation Decathlon and The Cancer Imaging Archive **The clinical data and the experimental details are provided in Sections 3.2 and 3.3 in the updated manuscript**.
>
>
> **Please address how the number of training rounds were determined. Wouldn't it show a more complete picture of the system if all agents were trained until convergence on a validation dataset?**
>
> The reviewer raises an important point. We determined the minimum number of training rounds as the number of training rounds necessary for all the agents in the system to be exposed to all the tasks within the system, whether through direct datasets or ERBs. Thus the minimum training rounds = 1+number of tasks/number of agents which resulted in three rounds for the BRATS dataset experiment.
>
> The goal of the ADFLL setup was to enable peer-to-peer experience sharing to learn new unseen tasks. As a result, we developed ADFLL with the hypothesis that different agents will only have a partial view of the complete validation set defined by the tasks they encounter and would not have access to the complete validation dataset that captures all the environments being trained in the framework.
>
> **Could the authors describe more technical details of what experience (eg. its dimensions) is shared between the agents?**
>
> The reviewer brought up a critical question about the clarification of ERBs. The ERB produced after DRL training is integral to our framework. An ERB contains sampled state-action-reward-resulting-state tuples that capture the DRL agents’ explorations of the image environment. The environment is represented by 3D imaging volume, and the agent was represented by a 3D bounding box capable of six distinct actions: moving positive or negative in the x, y, or z direction. The state was defined by an agent’s current location (or a chain of locations), each represented by a 3D bounding box of 45 × 45 × 11 pixels. The reward is calculated by the change in distance to the target landmark location before and after the agent takes action. The resulting state is the state of the agent after it takes an action. Thus the information in the ERBs is non-sensitive as the action and reward are information about the training session, and the state and resulting states are tiny fractions of the total 3D image, roughly 0.3%.

---

### Meta-Review · Area_Chair_hRJy · 2024-04-02

**Recommendation:** Accept (Poster)
**Confidence:** 4

**Metareview:**

All reviewers found the proposed method to be novel and the results promising. 3 weak accept and 1 borderline accept recommendations from reviewers.

---

### Decision · Program_Chairs · 2024-04-05

Accept (Poster)